# Cryo-electron tomography of NLRP3-activated ASC complexes reveals organelle co-localization

Yangci Liu[1,2], Haoming Zhai[1,2], Helen Alemayehu[1,2], Jérôme Boulanger[3], Lee J. Hopkins[4,5,6], Alicia C. Borgeaud[3,7], Christina Heroven[3,8], Jonathan D. Howe ®[3], Kendra E. Leigh ®[1,2], Clare E. Bryant ®[4,5] ✉ & Yorgo Modis ®[1,2] ✉

NLRP3 induces caspase-1-dependent pyroptotic cell death to drive inflammation. Aberrant activity of NLRP3 occurs in many human diseases. NLRP3 activation induces ASC polymerization into a single, micron-scale perinuclear punctum. Higher resolution imaging of this signaling platform is needed to understand how it induces pyroptosis. Here, we apply correlative cryo-light microscopy and cryo-electron tomography to visualize ASC/caspase-1 in NLRP3-activated cells. The puncta are composed of branched ASC filaments, with a tubular core formed by the pyrin domain. Ribosomes and Golgi-like or endosomal vesicles permeate the filament network, consistent with roles for these organelles in NLRP3 activation. Mitochondria are not associated with ASC but have outer-membrane discontinuities the same size as gasdermin D pores, consistent with our data showing gasdermin D associates with mitochondria and contributes to mitochondrial depolarization.

Inflammasomes are oligomeric signaling complexes that induce proteolytic activation of caspase-1, leading to processing of the proinflammatory cytokines interleukin (IL)−1β and IL-18, and cleavage of the cell death effector gasdermin D (GsdmD) to induce pyroptosis. In the first step of inflammasome assembly, Nod-like receptors (NLR) sense cytosolic chemical signatures associated with microbes and cell dysfunction or damage[1]. Ligand binding induces NLR oligomerization and recruitment of downstream proteins linked to the caspase-1 signaling cascade.

NLRP3 is one of the most studied inflammasome NLRs due to its association with many human diseases including degenerative disorders, dermatitis, and metabolic disorders[2,3]. Over 250 human NLRP3 protein variants have been identified, of which at least 30 have been associated with autoinflammatory disease through gain-of-function[4,5]. NLRP3 inflammasome activation has been linked to a broad spectrum of cellular processes or cytosolic chemical signatures including potassium efflux, reactive oxygen species, ribosomal arrest, calcium influx, chloride efflux, phosphatidylinositol-4-phosphate (PI4P) and trans-Golgi network (TGN) markers on endosomal membranes, and mitochondrial stress or damage[6–8]. In addition to ligand recognition, activation of the NLRP3 and pyrin inflammasomes may require transport along microtubules by dynein to the microtubule organizing center (MTOC)[9,10], with NLRP3 in an inactive cage-like oligomeric form[11,12]. Additionally, NLRP3 has also been reported to activate pyroptosis without colocalizing with TGN or MTOC markers and without forming cage-like oligomers[13]. Precisely how these different activation

[1]Molecular Immunity Unit, Department of Medicine, University of Cambridge, MRC Laboratory of Molecular Biology, Francis Crick Avenue, Cambridge CB2 0QH, UK. [2]Cambridge Institute of Therapeutic Immunology & Infectious Disease (CITIID), University of Cambridge School of Clinical Medicine, Cambridge CB2 0AW, UK. [3]MRC Laboratory of Molecular Biology, Francis Crick Avenue, Cambridge CB2 0QH, UK. [4]Department of Medicine, University of Cambridge, Box 157, Level 5, Addenbrooke's Hospital, Cambridge CB2 0QQ, UK. [5]Department of Veterinary Medicine, University of Cambridge, Madingley Road, Cambridge CB3 0ES, UK. [6]Present address: Wren Therapeutics, Clarendon House, Clarendon Road, Cambridge CB2 8FH, UK. [7]Present address: Institute of Biochemistry and Molecular Medicine, University of Bern, Bühlstrasse 28, 3012 Bern, Switzerland. [8]Present address: Division of Structural Biology, University of Oxford, Oxford OX3 7BN, UK. ✉e-mail: ceb27@cam.ac.uk; ymodis@mrc-lmb.cam.ac.uk

mechanisms and the associated involvement of subcellular organelles lead to NLRP3 activation remains to be elucidated.

Cryo-electron microscopy (cryo-EM) image reconstructions show that ligand-bound inflammasome NLRs assemble into disk- or split washer-like oligomers[14–17]. Upon activation, NLRP3 recruits the adaptor protein ASC (Apoptosis-associated speck-like protein containing a caspase recruitment domain (CARD)). NLRP3 and ASC interact through their pyrin domains (PYDs)[18]. ASC then can assemble into a single perinuclear punctum with a micron-scale diameter, also known as a speck, which functions as a platform for the recruitment and activation of caspase-1[19,20]. The PYD and CARD of ASC are both required for punctum formation and signaling[19,21,22]. Structural studies on purified ASC PYD and ASC CARD show that both domains have a death fold and independently form helical filaments[23–28]. In the cell, fluorescence microscopy imaging suggests that PYD-PYD interactions drive ASC filament formation whereas CARD-CARD interactions promote cross-linking and compaction of PYD filaments into puncta[21,22,27–30]. In addition, ASC recruits procaspase-1 via a CARD-CARD interaction between the two proteins[19,27]. Super-resolution light microscopy of endogenous ASC specks revealed punctate structures containing ASC, caspase 1 and NLRP3 within the same macroassembly[31]. These structures have yet to be imaged inside a cell at sufficient resolution to resolve the ASC or caspase-1 ultrastructure within puncta[30]. Hence, it remains unclear how the PYD and CARD contribute to the formation of an ASC punctum inside cells.

Understanding how ASC filament formation contributes to the assembly and activation of inflammasome puncta requires structural information in the cellular context. Recent advances in the preparation of frozen vitrified cellular lamellae by focused ion-beam (FIB) milling[32,33] have allowed the ultrastructure of multimeric protein complexes to be determined in a near-native context by cryo-electron tomography (cryo-ET). Here, we perform in situ cryo-ET to obtain three-dimensional image reconstructions of multimeric signaling complexes in cells with active NLRP3 inflammasomes. We identify ASC/caspase-1 signalosome puncta by correlative light and electron microscopy (CLEM) and reveal their ultrastructure and pyroptotic cellular landscape in immortalized bone marrow derived macrophages (iBMDMs) by cryo-ET image reconstruction. We find that inflammasome puncta are composed of short, hollow ASC/caspase-1 filaments, some branched, with variable packing density allowing ribosomes and trans-Golgi-lie vesicles to permeate the puncta. Neither the MTOC nor mitochondria are associated with the ASC speck although the ultrastructural analysis indicates local disruption or pore formation in the outer mitochondrial membrane. Time resolved analysis of NLRP3 stimulated ASC speck formation shows it occurs concurrently with loss of mitochondrial integrity. We propose that the ultrastructure of the ASC filament network functions as a signaling platform by providing structural integrity while allowing downstream signaling molecules to diffuse freely and bind at high density within the network.

## Results

### In situ cryo-CLEM of ASC/caspase-1 puncta

High-resolution structures of inflammasome components have been obtained from purified proteins. In a previous study in zebrafish larvae, electron tomography of ASC puncta in fixed larval sections stained with uranyl acetate showed that the puncta contained a dense filament network[30], but the imaging resolution was limited by chemical fixation and staining of the sample. To examine the architecture and cellular interactions of the NLRP3 signalosome in its physiological environment, we imaged ASC puncta in iBMDMs by correlative fluorescence light microscopy and in situ cryo-ET (cryo-CLEM). Cells expressing ASC fused with a C-terminal fluorescent protein have been used extensively to visualize ASC puncta formation[34]. In some experimental systems, ASC overexpression can lead to puncta formation in the absence of

activating inflammasome stimuli[34]. Fluorescence microscopy of live iBMDM cells overexpressing ASC-mCerulean used in this study showed that both priming with lipopolysaccharide (LPS) and stimulation with nigericin were still required to induce ASC puncta formation (Supplementary Fig. 1a). Time course analysis showed ASC speck formation occurring within 20 min of nigericin stimulation (Supplementary Movie 1). GsdmD cleavage observed by Western blotting and IL-1β cytokine release measured in ELISAs confirmed that the ASC-mCerulean puncta activated pyroptotic signaling (Supplementary Fig. 1b). Limiting the stimulation time with nigericin to 30 min allowed early-stage inflammasome signaling complexes to be captured, as indicated by colocalization of NLRP3, proteolytically activated caspase-1 and IL-1β with ASC in the puncta, in both wild-type (WT) iBMDMs and ASC-mCerulean iBMDMs (Supplementary Fig. 1c–e). NLRP3 and pro-IL-1β localization was determined by immunofluorescence. Caspase-1 localization was inferred using the carboxyfluorescein-labeled caspase-1 substrate FAM-FLICA. FAM-FLICA or its non-fluorescent analog Z-VAD-FMK also facilitated imaging by delaying pyroptotic cell death (Supplementary Movie 1).

For high-resolution imaging, cells grown on electron microscopy grids were vitrified by plunge-freezing in liquid ethane. We used both iBMDMs overexpressing ASC-mCerulean and WT iBMDMs labeled with FAM-FLICA after NLRP3 inflammasome activation[35]. ASC/caspase-1 signaling complexes were located by cryo-fluorescence light microscopy (cryo-FM; Supplementary Fig. 2a). Areas of the frozen cells containing an ASC or caspase punctum were FIB-milled down to lamellae 150–300 nm-thick and 15–20 μm wide[32,33], with micro-expansion joints to reduce bending[36] (Supplementary Fig. 2b, c). The lamellae were imaged by cryo-CLEM[37], using cryo-FM to locate any remaining fluorescent signal from ASC-mCerulean or FAM-FLICA in the lamellae after milling (Fig. 1a–c; Supplementary Figs. 2d, e and 3). In most cases the ASC punctum was milled away during the milling procedure but about 5% of lamellae retained fluorescent signal (Supplementary Fig. 3). The micron-scale dimensions of the fluorescent puncta allowed cryo-ET tilt-series acquisition at regions of interest with the necessary targeting precision after correlation of the cryo-FM images with low-magnification cryo-EM maps (Supplementary Fig. 2f, g)[38,39]. To obtain an internal quality metric for the resulting tomograms, we generated a subtomogram averaging reconstruction of 1058 ribosomes extracted from five ASC speck tomograms. This yielded a ribosome structure with a resolution of 23 Å and the expected structural features (Fig. 1e, f; Supplementary Fig. 2h, j).

### Composition, ultrastructure, and dynamics of ASC/caspase-1 puncta

Three-dimensional cryo-ET image reconstructions show a dense network of branched filaments in areas identified by cryo-CLEM as containing ASC-mCerulean or caspase-1 labeled with FAM-FLICA (Fig. 1d; Supplementary Fig. 2g). A three-dimensional model of the ultrastructure generated with IMOD[40] showed that ASC puncta consist of several core regions of densely packed filaments separated by sparser regions, with branching filaments connecting adjacent densely packed regions (Fig. 1e, f; Supplementary Movie 2). Ribosomes were abundant within the filament network. Small vesicles were present, but organelles including mitochondria and the MTOC were excluded. A typical cryo-ET reconstruction (1900 × 1400 × 108 nm) contained 10–20 ribosomes in contact with ASC filaments and 2–4 sites with filaments within 15 nm of endoplasmic reticulum (ER) or vesicle membranes (Fig. 1e, f). Filaments in cells expressing ASC-mCerulean had additional density at the filament periphery, and a less distinct outline compared to filaments in cells expressing WT ASC (Fig. 2a). This suggests the additional density is primarily attributable to mCerulean, which at 27 kDa is larger than ASC (22 kDa), although other components such as caspase-1 (which was shown by light microscopy to localize to ASC specks), may also contribute to this density. A green fluorescent

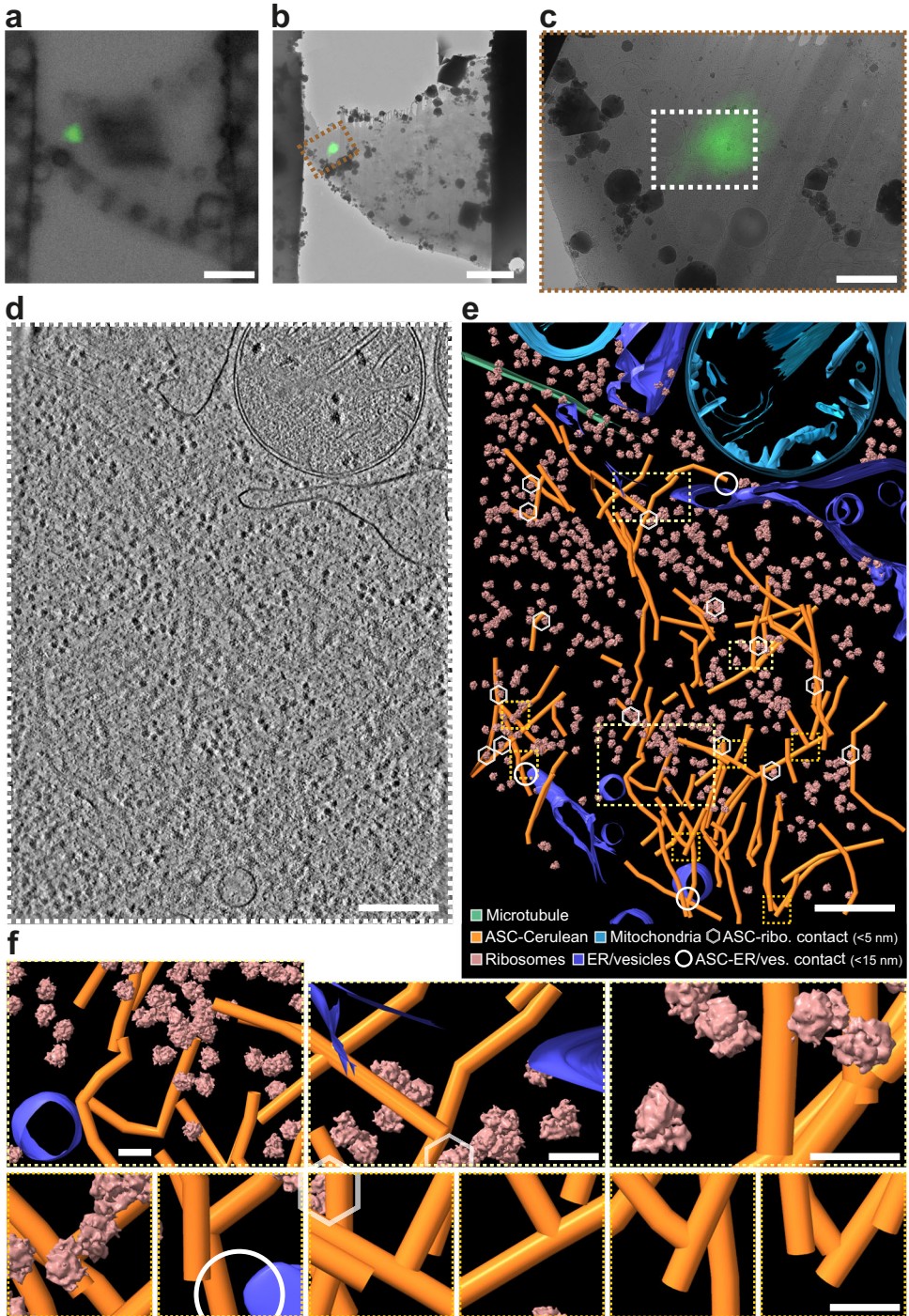

**Fig. 1 | Cryo-ET image reconstructions and models of ASC-caspase 1 puncta in iBMDMs. a** Cryo-FM image of a lamella from an iBMDM expressing ASC-mCerulean. ASC-mCerulean fluorescence is shown (green), overlaid on a bright-field image. Representative image of four biological replicates. Scale bar, 5 μm. **b** Cryo-FM image shown in (**a**) correlated with and overlaid on a cryo-EM map of the lamella with eC-CLEM ICY[37]. Representative image of four biological replicates. Scale bar, 5 μm. **c** Closeup of the area boxed in red in (**b**). Scale bar, 1 μm. **d** Reconstructed cryo-ET volume (13.6-nm thick virtual tomographic slice) of the area boxed in white in (**c**). Scale bar, 250 nm. **e** 3-D segmented model of an 1873 × 1386 × 108 nm volume covering the area shown in (**d**), generated with IMOD[40]. Scale bar, 250 nm. Hexagons denote ASC-ribosome contacts (<5 nm). Circles denote ASC-ER/vesicle contacts (<15 nm). **f** Closeups of the boxed areas in (**e**): Yellow boxes, representative views at different magnifications. Orange boxes, ASC filament branch points. Scale bars, 40 nm. The ribosome structure shown is a 23-Å resolution subtomogram averaging reconstruction of 1058 ribosomes from five tomograms of ASC-mCerulean puncta (see Supplementary Fig. 2).

protein tag was similarly found to decorate cryo-ET reconstructions of amyloid-like poly-Gly-Ala aggregates with additional densities[41]. Since mCerulean was fused to the C-terminus of ASC, the presence of mCerulean at the filament periphery supports a model in which the filaments are formed at their core by the N-terminal PYD of ASC, with

the CARD located in between the PYD and mCerulean. Indeed, the cryo-EM structure of purified recombinant ASC PYD filaments and the NMR structure of full-length ASC, taken together, suggest that the PYD forms the filament core (9-nm in diameter), with the flexibly-linked CARD increasing the filament diameter to 16−18 nm[27,28]. In our cryo-ET

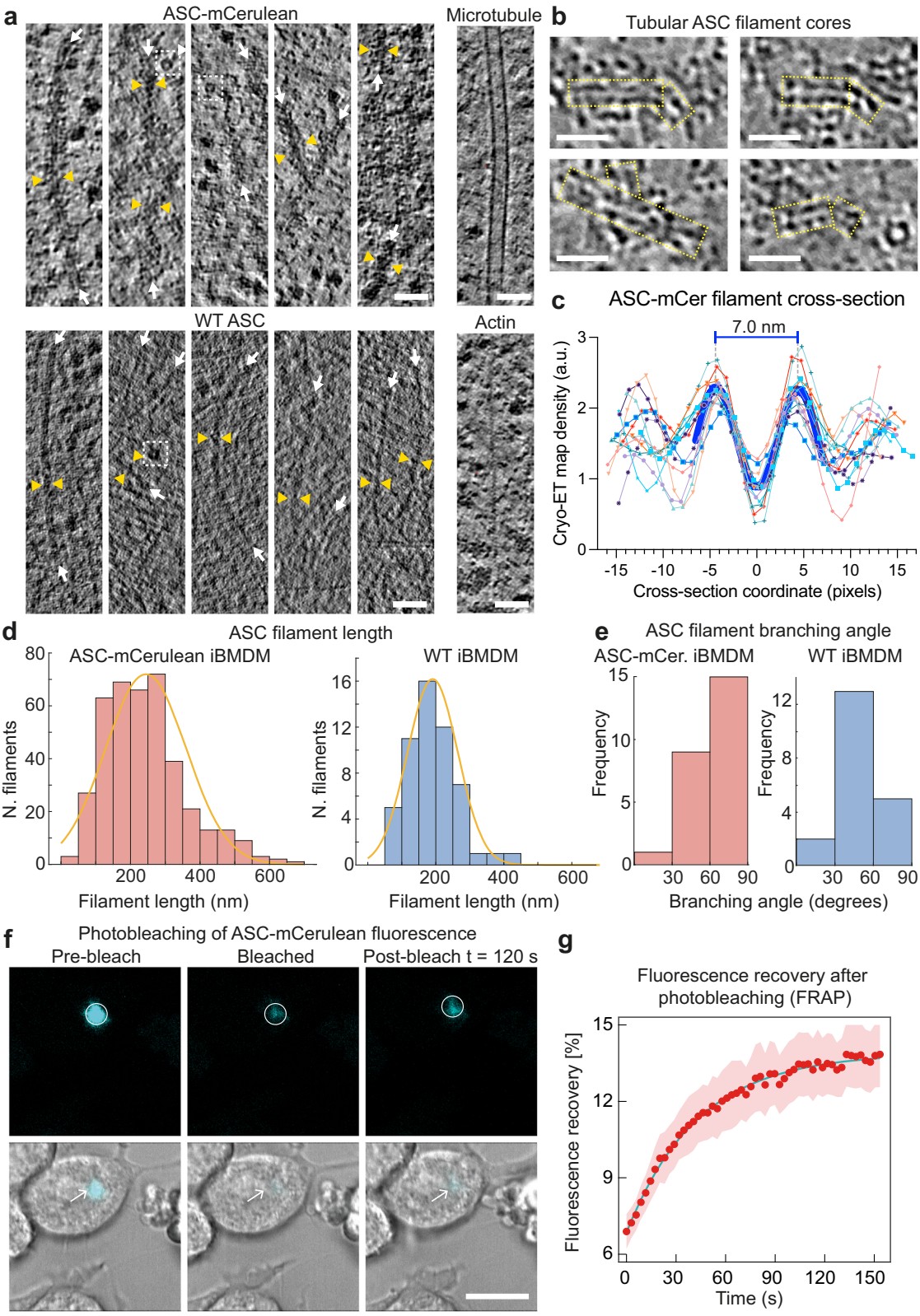

reconstructions, filaments containing WT ASC had a diameter of 11–15 nm, which increased to 22–30 nm for filaments containing ASC-mCerulean (Fig. 2a). Notably, a hollow tubular core of density 7 nm in diameter (Fig. 2b, c, and Supplementary Fig. 4) was visible within the ASC-mCerulean filaments, for which the cryo-ET images were of higher quality. Neural network image restoration with cryoCARE[42] enhanced the definition of the tubular filament core. Although fluorescence from

ASC-mCerulean and FAM-FLICA-labeled capsase-1 colocalized in iBMDMs (Supplementary Fig. 1e), the resolution of cryo-CLEM was insufficient to determine whether any of the cryo-ET filament densities corresponded to caspase-1 or other components. We note, however, that caspase-1 (30 kDa) and procaspase-1 (45 kDa) are both larger than ASC and would therefore be expected to increase filament diameter significantly, if present with the same stoichiometry as ASC.

**Fig. 2 | Ultrastructure and dynamics of the ASC filament network in puncta in iBMDMs. a** Filaments in ASC-mCerulean puncta (upper left), or FAM-FLICA-labeled caspase-1 puncta, (lower left), in 4-nm thick virtual tomographic slices. White arrows indicate filament axes. Pairs of yellow triangles show filament thickness. Dashed white boxes highlight ring-shaped densities. Microtubules and actin filaments from the ASC-mCerulean tomogram are shown for reference. Scale bars, 50 nm. **b** Tubular filament cores (boxed in yellow) in representative 8-Å thick virtual tomographic slices after image restoration with cryoCARE[42]. Scale bars, 25 nm. **c** Cryo-ET density profiles from 13 ASC filament cross-section areas in 8-Å tomographic slices. 1 pixel = 8 Å. See Supplementary Fig. 4 for details on how density profiles were plotted. **d** Filament length distribution in ASC-mCerulean iBMDMs (left) and WT iBMDMs labeled with FAM-FLICA (right). **e** Distribution of branching angles in ASC-mCerulean puncta (from 2 tomograms), and a WT ASC punctum (1 tomogram). **f** Fluorescence recovery after photobleaching (FRAP) of ASC-mCerulean. Scale bar, 10 μm. Two independent experiments were performed. A total of eight cells were imaged for puncta formation and FRAP. **g** FRAP curve used to calculate the ASC-mCerulean dissociation rate ($k_{off}$) from the bleached area. Shaded area represents ± s.e.m. for eight imaged cells (n = 8). See Source Data File for source data.

Filament length measurements showed that WT filaments were on average slightly shorter (191 ± 74 nm, $n$ = 54 from one tomogram) than ASC-mCerulean filaments (245 ± 116 nm, $n$ = 402 from six tomograms), with similarly broad length distributions (Fig. 2d). This difference could potentially be due to the higher levels of ASC protein expression in iBMDMs expressing ASC-mCerulean, which also express endogenous WT ASC. Analysis of the branching angles showed that the angle distributions of untagged and tagged filaments were slightly different, with larger branching angles more common in mCerulean-tagged filaments, potentially due to steric constraints from the tag (Figs. 1f, 2e, and Supplementary Fig. 4). Our filament branching angle measurements are likely underestimated as any filaments with their axis closely aligned with the Z-axis−which would result in a large (70−90˚) branching angle−would not have been detected due to the missing-wedge effect (from sample-tilt limitations). Taken together with previously published structural data from purified proteins, our cryo-ET reconstructions suggest ASC filaments form the backbone of inflammasome-induced puncta, but that the branching of these filaments results in a structural scaffold that can recruit and concentrate downstream effector proteins.

We next investigated the equilibrium dynamics of ASC-mCerulean filaments by fluorescence recovery after photobleaching (FRAP) in iBMDMs. We bleached circular regions containing ASC-mCerulean puncta and imaged the ASC-mCerulean fluorescence intensity before and after bleaching (Fig. 2f). Our data showed that ASC-mCerulean filament assemblies have a small mobile fraction (6.9%; Fig. 2g). The dissociation rate ($k_{off}$) of ASC-mCerulean from the imaged area was measured from the rate of fluorescence recovery as 0.023 ± 0.001 s$^{-1}$ (Fig. 2f, g). Most of the observed fluorescence recovery was at the periphery of puncta, suggesting that the ASC-mCerulean filaments inside the puncta are largely immobile and are not in equilibrium with ASC monomers or oligomers in the cytosol (Fig. 2f).

## Co-localization of ceramide-rich vesicles with ASC specks
How NLRP3 is activated by a broad range of different stimuli – including ion fluxes, reactive oxygen species and mitochondrial damage – remains an open question. A common feature of NLRP3 activation by these different stimuli is that ER-endosome contact sites are disrupted, thereby disrupting trafficking from endosomes to the *trans*-Golgi network (TGN). This in turn causes accumulation of PI4P and TGN markers including TGN38 in scattered endosomes, on which NLRP3 accumulates[7]. These NLRP3 foci have been proposed to serve as nucleation sites for ASC and caspase-1 recruitment and activation[8,9]. Fluorescent ceramide analogs such as BODIPY TR ceramide selectively accumulate in the Golgi[43]. We monitored localization of BODIPY TR ceramide during ASC speck formation, using either ASC-mCerulean or FAM-FLICA-labeled capsase-1 fluorescence to visualize the specks in live iBMDMs. BODIPY TR ceramide fluorescence was consistently enriched at sites of speck formation (Fig. 3a; Supplementary Movie 3). Immunofluorescence of TGN38 showed that this TGN marker was present in foci dispersed throughout the cytosol of WT iBMDMs, ASC-mCerulean IBMDMs, and THP-1 cells (Fig. 3b, c; Supplementary Fig. 5b), consistent with the reported localization of TGN38 in scattered endosomal vesicles[7]. Moreover, NLRP3 substantially colocalized

with TGN38 in WT iBMDMs primed with LPS, with or without nigericin stimulation (Fig. 3b, Supplementary Movie 4). BODIPY TR ceramide also partially colocalized with TGN38 (Supplementary Fig. 5a). However, there was little direct overlap between ASC-mCerulean and anti-TGN38 fluorescence (Fig. 3c). In our tomographic reconstructions, we observed vesicles with diameters of 50−300 nm within the ASC filament network (Fig. 3d, e; Supplementary Figs. 2g and 5c; Supplementary Movie 2). We observed similar vesicles within ASC-mCerulean puncta in chemically fixed samples imaged by room temperature electron tomography (Supplementary Fig. 5d−f). The presence of ceramide and absence of TGN38 in the vesicles within ASC puncta suggests these vesicles are either *cis*-Golgi vesicles or endosomes in which ceramides have accumulated. Ribosomes were also found embedded throughout the ASC filament network. Ring-shaped densities were occasionally observed within tomograms of ASC puncta, but the resolution of the reconstructions was insufficient to infer the composition of these features (Fig. 2a). Together, our cryo-ET and fluorescence microscopy data supports the model that the vesicles enriched in ceramides may serve as a platform for ASC recruitment to multiple NLRP3 foci, which nucleate ASC polymerization, with subsequent branching of ASC filaments driving growth into a micron-scale punctum.

Since NLRP3 inflammasome activation was reported to require transport of NLRP3 to the MTOC[9,10], we examined the localization of ASC/caspase-1 puncta relative to the MTOC. Fluorescent markers for ASC and caspase-1 did not colocalize with the MTOC components γ-tubulin, pericentrin or ninein in WT or ASC-mCerulean iBMDMs stimulated with LPS and nigericin (Supplementary Fig. 6). In stimulated THP-1 cells, ASC/caspase-1 puncta were on average closer to the MTOC, with one quarter of puncta within 2 μm of the MTOC, but there was little direct overlap between the fluorescent markers (Supplementary Fig. 6). This is consistent with a report of an NLRP3 activation pathway that does not involve colocalization of NLRP3 with MTOC markers[13].

## Aberrant mitochondrial morphology during pyroptosis visualized by cryo-ET
Mitochondria function as a nexus for multiple pathogen-sensing and damage-sensing signaling pathways. Innate immune sensors of viral nucleic acids[44] and mitochondrial damage converge on the outer mitochondrial membrane, and induce apoptosis or pyroptosis, depending on the specific danger signal that is sensed[45,46]. To visualize and analyze the morphology of mitochondria soon after inflammasome activation we performed cryo-ET reconstructions of mitochondria following LPS priming and nigericin stimulation to induce ASC speck formation. In LPS primed iBMDMs and THP-1 cells expressing WT or ASC-mCerulean ASC, stimulation with nigericin resulted after 30−45 min in mitochondria with a more rounded overall shape and smaller size than in unstimulated cells (Fig. 4a, and Supplementary Fig. 7). Cryo-ET reconstructions of iBMDMs and THP-1 cells 1 h post-stimulation showed that the mitochondria in these cells had an abnormal ultrastructure, with the inner membranes forming concentric tubular structures instead of the regularly spaced lamellar cristae observed in unstimulated cells (Fig. 4a, and Supplementary

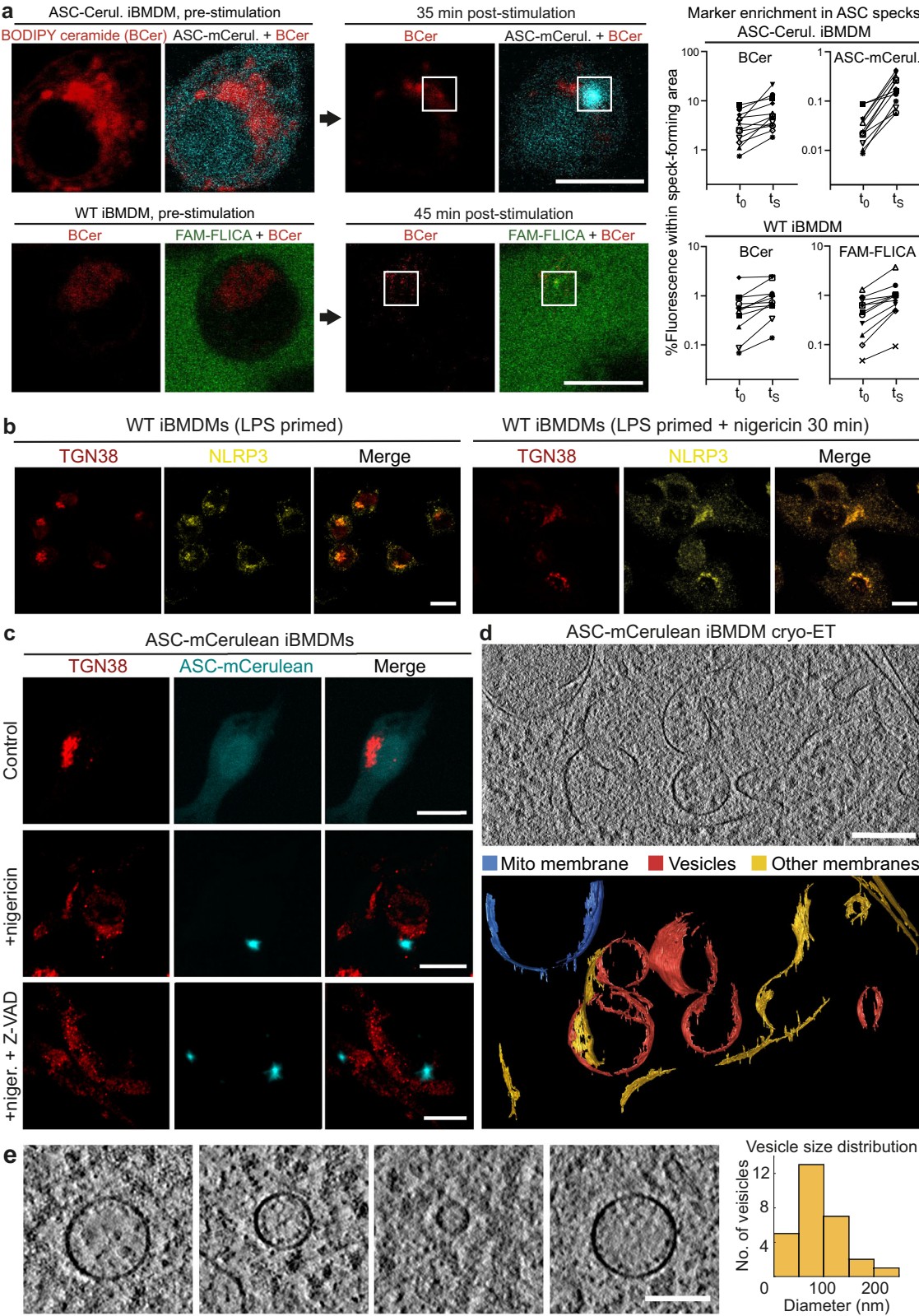

Fig. 7). To quantify the changes in mitochondrial morphology associated with ASC-dependent signaling, four parameters were extracted and measured using Fiji:[47] cristae spacing, inner-to-outer membrane spacing, cristae lumen width and cristae apex angles (Fig. 4b, c). We found the cristae spacing to be two- to fourfold smaller in pyroptotic mitochondria than in healthy mitochondria. The other three parameters were not significantly different.

## Protein-rich pores 10–20 nm in diameter in the outer mitochondrial membrane

During apoptosis, the Bax and Bak proteins create large disruptions in the outer mitochondrial membrane[45,48]. Our cryo-ET reconstructions showed that mitochondria in pyroptotic iBMDMs stimulated with LPS/nigericin lacked any such large outer membrane disruptions. We did, however, identify smaller discontinuities in the outer membranes of

**Fig. 3 | Localization of vesicles with accumulated ceramides in ASC specks by live-cell fluorescence confocal microscopy and cryo-ET. a** Distribution of BODIPY TR ceramide (BCer) during ASC speck formation in iBMDMs expressing ASC-mCerulean, or WT iBMDMs stained with FAM-FLICA. Areas of speck formation are boxed. See Supplementary Movie 3 for full time courses. Right, percentage of whole-cell fluorescence that mapped to the area of speck formation, before and after speck formation, $t_0$ and $t_S$ (1–3 min and 20–50 min after nigericin addition), respectively. 12 specks from ASC-mCerulean iBMDMs and 11 specks from WT iBMDMs were analyzed. See Source Data File for source data. Scale bars, 10 μm. **b** Immunofluorescence microscopy of LPS-primed WT iBMDMs with or without nigericin stimulation. Anti-NLRP3 partially colocalized with anti-TGN38. The TGN was dispersed following stimulation. Scale bar, 10 μm. See Supplementary Movie 4 for a Z-stack series. The images are representative of three independent experiments, with at least 20 cells imaged per experiment. **c** Immunofluorescence microscopy of LPS-primed ASC-mCerulean iBMDMs with or without nigericin stimulation, and with nigericin stimulation and caspase inhibitor Z-VAD-FMK. There is little overlap between ASC-mCerulean and anti-TGN38 fluorescence. Scale bars, 10 μm. The images are representative of three independent experiments, with at least 20 cells imaged per experiment. **d** 1.36-nm thick virtual tomographic slice within an ASC speck. Scale bar, 200 nm. Lower panel, 3-D segmentation model. **e** Vesicles from cryo-ET reconstructions of ASC-mCerulean iBMDMs. Scale bar, 100 nm. The histogram shows the vesicle size distribution.

some mitochondria in the stimulated ASC-mCerulean and WT iBMDMs (Fig. 5a and Supplementary Fig. 8). The size of these gaps, which were only present after nigericin stimulation, varied from 10 to 20 nm (Fig. 5b and Supplementary Fig. 9a). The inner membrane was intact near outer membrane gap sites, implying that the gaps were not due to the missing wedge effect or the anisotropic resolution of cryo-ET tomograms[49].

The cryo-ET map density within outer membrane gaps was lower than that of lipid membranes but higher than in the intermembrane space (Fig. 5b). The cryo-ET density at gap sites was greater than in the adjacent intermembrane space over a span of 10–20 nm along the Z-axis, similar to the dimensions of the gaps in X and Y (Supplementary Fig. 9a). This suggests the gap sites contain a higher concentration of protein than the intermembrane space. We conclude that protein-rich gaps or pores, 10–20 nm in diameter, form in the outer mitochondrial membrane during pyroptosis.

### GsdmD pore-forming domain inserts into mitochondria and contributes depolarization

To determine how these outer membrane gaps may contribute to mitochondrial outer membrane permeabilization (MOMP), we quantified membrane potential following inflammasome stimulation in iBMDMs by flow cytometry, using tetramethylrhodamine (TMRM) fluorescence as a reporter. Stimulation with LPS and nigericin or LPS and ATP caused loss of mitochondrial membrane potential in a large fraction of cells (40–50%; Fig. 5c and Supplementary Fig. 7).

The N-terminal domains of gasdermins D and E form 20-nm pores in lipid bilayers[50,51], and have been reported to cause mitochondrial depolarization and DNA release upon inflammasome activation[52–55]. To assess the role of GsdmD in mitochondrial depolarization under the conditions used for this cryo-ET study, we measured the mitochondrial membrane potential of GsdmD[-/-] iBMDMs upon inflammasome stimulation. Stimulation of GsdmD[-/-] cells with LPS and nigericin induced depolarization in only 23% of cells (versus 39–48% for WT; Fig. 5c, d, and Supplementary Fig. 10a, b). Moreover, when LPS and ATP were used for stimulation, WT and GsdmD[-/-] cells were depolarized to the same extent (Supplementary Fig. 10d). In contrast, knockout of Ninj1, which promotes plasma membrane rupture during pyroptosis[56], increased depolarization with either nigericin or ATP stimulation (Supplementary Fig. 10c, d). This suggests that GsdmD contributes to mitochondrial depolarization in cells stimulated with nigericin in a manner independent of plasma membrane rupture. We hypothesized that the GsdmD-dependent component of mitochondrial depolarization could be due to GsdmD pores forming in the outer mitochondrial membrane. Consistent with this, subcellular fractionation experiments showed that upon stimulation with LPS and nigericin GsdmD was proteolytically cleaved and the pore-forming N-terminal proteolytic cleavage fragment was translocated into a purified mitochondrial fraction (Fig. 5e and Supplementary Fig. 9b). The outer mitochondrial membrane protein Tom20 was enriched in the purified mitochondrial fraction as expected, but notably, the plasma membrane receptor CD14 was absent in the mitochondrial fraction, indicating that the mitochondrial fraction did not contain any contaminants from the plasma membrane. Cryo-EM images of the purified mitochondrial fraction confirmed that it contained mitochondrial with key morphological features largely preserved (Supplementary Fig. 9c).

### Discussion

The formation of NLRP3 inflammasome signaling assemblies into puncta, or specks, containing ASC and caspase-1 is a cardinal feature of inflammasome activation. Here, our cryo-ET reconstructions, obtained in unstained and fully hydrated conditions, show that the speck is formed of a filamentous network consisting of hollow-tube branched filaments with the dimensions predicted for ASC filaments, based on structural studies of purified ASC PYD filaments and full-length monomeric ASC[27,28]. The identification of a filamentous network containing mCerulean-labeled ASC, similar to an ASC punctum seen in a fixed zebrafish section[30], positively identifies this protein as the principal filament-forming component in the puncta. These data are supported by analysis of similar structures in wild-type cells labeled with a fluorescent caspase 1 substrate. The filament structure of the ASC with an mCerulean tag on the C-terminus supports the model whereby the PYD forms the filament core, and the CARD decorates the filament. The filament branching visible in the reconstructions is proposed to depend on CARD-CARD interactions because the CARD is required for puncta formation, whereas the ASC PYD alone forms unbranched filaments that lack the structural stability that is inherent to a branched network[21,22,27–30].

The density of ASC filaments varied in the cryo-ET reconstructions, with multiple core regions of densely packed filaments separated by sparser regions. This suggests that the higher-order structure of the punctum was likely seeded by multiple oligomeric assemblies, rather than growing concentrically from a single nucleation site. In contrast with huntingtin and other neurotoxic protein aggregates, which exclude other macromolecules based on cryo-ET reconstructions[57], the structural organization of the ASC filament network allows ribosomes and small Golgi-like vesicles to be retained in the puncta, or to permeate through them. Permeability of the ASC network to macromolecules could be important for efficient recruitment and release of effector, substrate, and product molecules. Permeability to vesicles would allow active transport of Golgi vesicles carrying activated inflammasome seed-oligomers from sites of NLRP3 activation to the site of punctum formation, as recently proposed[9]. Moreover, the ribosomes within the puncta could potentially contribute to the inflammasome signaling program, for example by expressing proteins required for signaling, or by responding to danger signals. Indeed, NLRP3 signaling can be activated by translational arrest through direct and indirect mechanisms[58], including binding of fungal polysaccharides to ribosomes[59].

Our cryo-ET reconstructions, in addition to revealing the ultrastructure of ASC within the cell, also allowed morphometric analysis of mitochondria in the inflammasome activated cellular environment. The clearest differences between mitochondria in inflammasome activated cells and unstimulated cells were a two to fourfold reduction

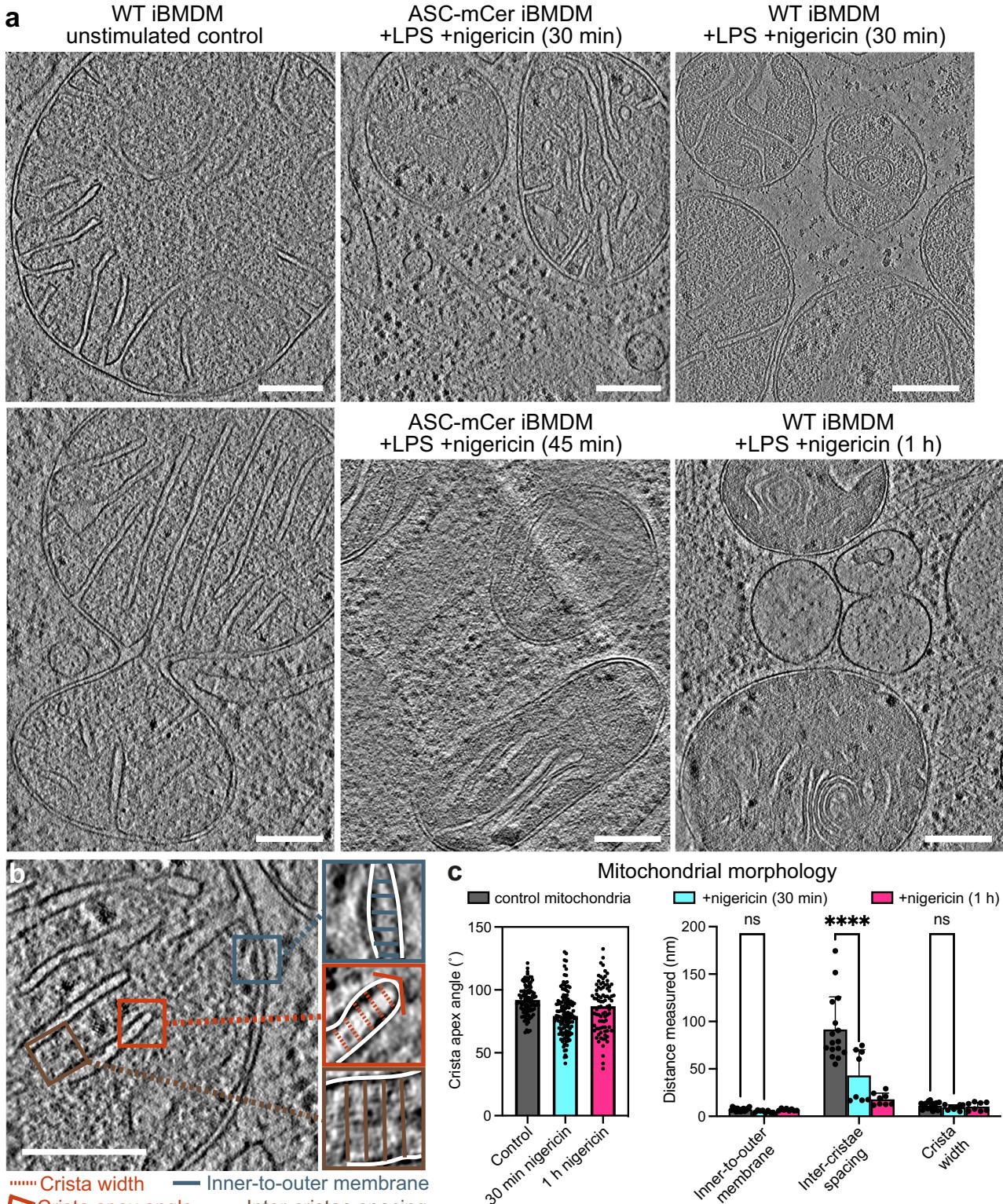

**Fig. 4 | Mitochondrial morphology before and after NLRP3 activation.**
**a** Reconstructed cryo-ET tomographic slices of iBMDMs expressing WT or mCerulean-labeled ASC at different timepoints after stimulation with nigericin. See Supplementary Fig. 7 for additional examples including from THP-1 cells. Scale bars, 200 nm. **b** Cryo-ET reconstruction of a mitochondrion in an unstimulated WT iBMDM with closeup panels defining the following morphological parameters: inner-to-outer membrane spacing, inter-cristae spacing, crista width and crista apex angle. Scale bar, 200 nm. **c** Quantitative analysis of the morphological

parameters defined in (**b**) in segmentation models of the mitochondrial membranes, as shown in (**a**). Parameters were measured in four tomograms for the control and two tomograms for each of the nigericin-stimulated samples. Error bars represent standard deviation from the mean. For the crista apex angles: "control", $n = 327$; "30 min", $n = 187$; "1 h", $n = 96$. For other parameters: "control", $n = 16$; "30 min", $n = 8$; "1 h", $n = 8$. Statistical test: two-way ANOVA, $P = 10^{-4}$ (****). See Source Data File for source data.

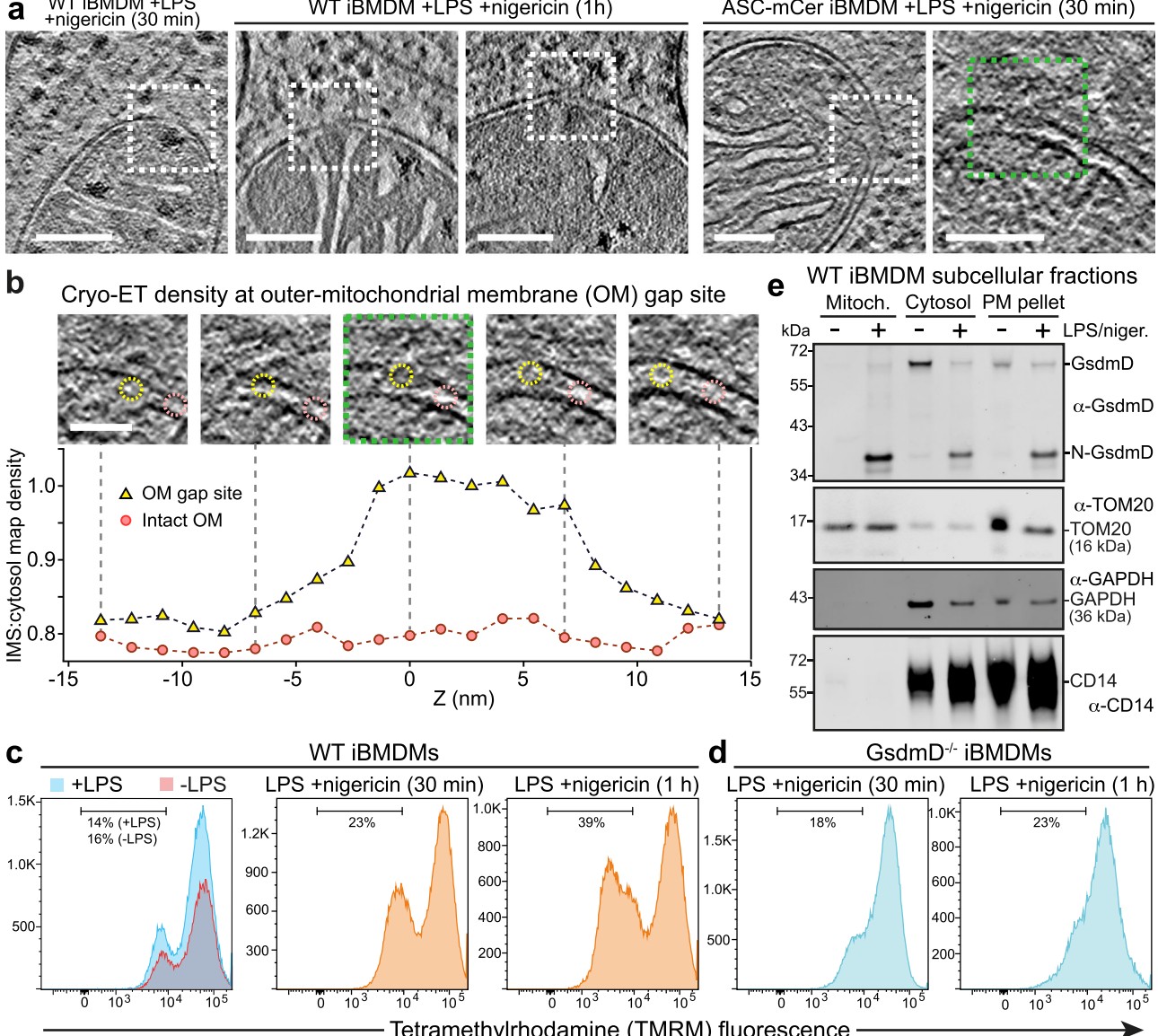

**Fig. 5 | Mitochondrial pore formation, GsdmD-dependent depolarization, GsmdD cleavage and mitochondrial association of N-GsdmD in NLRP3-activated cells.** Examples of mitochondria with discontinuities in the outer membrane (OM) in cryo-ET reconstructions of in ASC-mCerulean iBMDMs stimulated with LPS and nigericin. Dashed boxes denote the OM discontinuities. Scale bars, 100 nm. See Supplementary Fig. 8 for additional examples of OM discontinuities. Z-stack series and cryo-ET density measurements for the OM gap boxed in green in (**a**). Scale bar, 50 nm. The graph shows the cryo-ET density at the OM gap site (yellow) and at an adjacent site with intact inner and outer membranes (pink). Cryo-ET density is expressed as the density in the intermembrane space (IMS) divided by the density in a nearby cytosolic area. The areas in which IMS density was measured are circled in the Z-stack panels (yellow, gap site; pink, intact site). See Source Data File for source data. Flow cytometry histograms of WT iBMDMs, (**c**), or GsdmD[-/-] iBMDMs, (**d**), stained with tetramethylrhodamine (TMRM), a mitochondrial membrane potential reporter. See Supplementary Fig. 10 for additional controls. Vertical axes indicate cell count. **e** Immunoblots of subcellular fractionation of WT iBMDMs 60 min after stimulation with LPS and nigericin. The GsdmD N-terminal domain (N-GsdmD) is enriched in the mitochondrial fraction. Shown below are immunoblots for Tom20 (a mitochondrial protein), GAPDH (a cytosolic protein), and CD14 (a plasma membrane protein). See also Supplementary Fig. 9. The blots are representative of four independent experiments with similar results.

in cristae spacing, and the appearance of protein-rich discontinuities of 10–20 nm in diameter in the outer membrane, which are distinct from the larger BAK/BAX macropores[45,48]. Permeabilization of both the plasma membrane and outer mitochondrial membrane was reported previously in cells undergoing caspase-1-dependent pyroptosis after inflammasome activation[46]. Pore formation in the plasma membrane by the N-terminal domain of GsdmD after cleavage by caspase-1 is required for inflammasome-dependent pyroptosis[50] and gasdermin D and E pores have been reported to form on the outer mitochondrial membrane following inflammasome activation[52–55]. The outer-membrane discontinuities in the mitochondria of inflammasome-

activated cells had similar dimensions to GsdmD pores, which have an average inner diameter of 22 nm[50] (or 13–34 nm based on atomic force microscopy[51]). We show here that there is a GsdmD-dependent component to the depolarization mitochondrial membranes in iBMDMs undergoing pyroptosis following stimulation with LPS and nigericin. Moreover, we show that GsdmD is cleaved and the pore-forming N-terminal fragment translocates into the mitochondrial subcellular fraction upon NLPR3 activation. We conclude that GsdmD directly contributes to mitochondrial depolarization during pyroptosis by inserting and forming pores in the outer mitochondrial membrane. Stimulation with ATP instead of nigericin leads to more rapid and

extensive loss of mitochondrial potential (Supplementary Fig. 10). The same extensive ATP-induced depolarization occurs in GsdmD[-/-] and Ninj1[-/-] cells, suggesting that nigericin and ATP cause depolarization via distinct mechanisms. Further studies are warranted to determine how stimulation with extracellular ATP induces depolarization independent of GsdmD and Ninj1-mediated membrane rupture.

Overall, our structural analyses suggest that the filament branching and packing density within ASC puncta provide structural integrity while allowing downstream signaling molecules to diffuse freely and bind at high density within the network. Our cryo-ET reconstructions provide direct visualization of the cellular organelles (ribosomes and Golgi vesicles) within the ASC speck at a sufficient resolution to support their potential roles in NLRP3 activation. Although many structural details of the NLR inflammasome signaling machinery remain unclear, this study demonstrates the potential for combined cryo-CLEM and cryo-ET approaches to extract detailed, hypothesis-generating ultrastructural information for fully assembled innate immune signaling complexes in their cellular context.

## Methods

### Mammalian cell culture

Wild-type immortalized primary mouse bone-marrow derived macrophages (WT iBMDMs) and ASC-mCerulean iBMDMs, kindly provided by Eicke Latz (Univ. Bonn), were cultured in Dulbecco's modified Eagle Medium high glucose (DMEM; Gibco), supplemented with 10% v/v fetal bovine serum (FBS; Gibco). Human monocytic THP-1 cells (European Collection of Authenticated Cell Cultures) were grown in Roswell Park Memorial Institute (RPMI) 1640 Medium (Gibco), supplemented with heat inactivated 10% v/v FBS, 10 mM HEPES pH 7.4, 1 mM Sodium Pyruvate and 0.05 β-mercaptoethanol. Cells were checked regularly for mycoplasma contamination by MycoAlert™ mycoplasma detection kit (Lonza) and cells were free of mycoplasma contamination.

### Inflammasome stimulation

For stimulation of WT iBMDMs and ASC-mCerulean iBMDMs, cells were primed with 200 ng/ml LPS for 3 h in Opti-MEM® (Gibco) or DMEM supplemented with 10% FBS before NLRP3 stimulus was used. The following stimulus were used: 10 µM nigericin (Sigma or Invivogen) and 5 mM ATP (pH adjusted to 7.4; Sigma) for 30 min or 60 min with pan-caspase inhibitor Z-VAD-FMK (Invivogen) or carboxyfluorescein-labeled inhibitor of caspase-1 YVAD (FAM-FLICA; ImmunoChemistry Technologies) for in situ cryo-ET sample preparation unless otherwise specified. For stimulation of THP-1 cells, priming and stimulation were performed in RPMI Medium and conditions were listed above.

### Cytokine measurement and immunoblotting

IL-1β secretion was measured by ELISA following a previously established protocol described[31]. Supernatants were collected after inflammasome activation and the OptEIA kit (BD BioSciences) was used to measure mouse IL-1β concentration in cell supernatant according to the manufacturer's instructions. For the immunoblot in Supplementary Fig. 1, ASC-mCerulean iBMDMs were lysed in cell lysis buffer (150 mM NaCl, 50 mM Tris-HCl pH 8, 1% Triton X-100, 1 mM PMSF, 10 µg/mL leupeptin, 1 µg/mL aprotinin) on ice for 10 min before centrifugation. Supernatant was collected and boiled in SDS sample loading buffer for 5 min. Following separation by 4–20% gradient SDS-PAGE, the proteins were transferred to a PVDF membrane. Membrane was blocked in 5% milk in PBS + 0.2% Tween-20. The primary antibody was rabbit monoclonal anti-mouse GsdmD [EPR19828] (Abcam, ab209845, RRID:AB_2783550)], 1:1,000 dilution. The secondary antibody was goat anti rabbit IgG-HRP (Santa Cruz Biotechnology, sc-2004, RRID:AB_631746), 1:1000 dilution.

### Immunofluorescence staining

For immunofluorescence, cells were plated as monolayer on µ-Slide 8-Well chamber (ibiTreat, Ibidi, 80826) or 12-well chamber (removable, Ibidi, 81201). Cells were primed and stimulated as above, and then fixed with 4% paraformaldehyde for 5–10 min at room temperature. Cells were permeabilised and blocked in 0.1% Saponin (Sigma, 47036) supplemented with 20% FBS (Gibco) or 2% Bovine Serum Albumin (Sigma, A7030) in PBS with primary antibody overnight. Washing was performed before cells were incubated with secondary antibodies. The following primary antibodies were used: rabbit anti-ASC rabbit polyclonal pAb AL177, 1:200 dilution (AdipoGen, AG-25B-0006, RRID:AB_2490440); rabbit anti-ASC monoclonal ASC/TMS1(D2W8U), 1:800 dilution (Cell Signaling Technology, 67824, RRID:AB_2799736); goat anti-NLRP3 polyclonal, 1:200 dilution (Abcam, ab4207, RRID:AB_955792); goat anti-IL-1β polyclonal, 1:500 dilution (R&D Systems, AF-401-NA, RRID:AB_416684); mouse anti-γ-tubulin monoclonal, 1:400 dilution (Sigma-Aldrich, T6557, RRID:AB_477584); rabbit anti-TGN38 polyclonal, 1:250 dilution (Novus Biologicals, NBP1-03495, RRID:AB_1522533); rabbit anti-TOM20 FL-145 polyclonal, 1:80 dilution (Santa Cruz Biotechnology, sc-11415, RRID:AB_2207533). The Alexa Fluor secondary antibodies (488, 555, 568 and 647) used were: 488-labeled donkey anti-goat IgG (H + L) (1:500, ThermoFisher, A11055, RRID:AB_2534102); 555-labeled goat anti-rabbit IgG(H + L) (1:500, ThermoFisher, A21428 RRID:AB_141784); 568-labeled goat anti-mouse IgG (H + L) (1:500, ThermoFisher, A11004, RRID:AB_2534072); 647-labeled goat anti-rabbit IgG (H + L) (1:500, ThermoFisher, A21244, RRID:AB_2535812). After incubation with secondary antibodies at room temperature for 1 h, cells were washed three times with blocking buffer, one time with PBS and final wash with water before mounting with ProLong Gold Antifade Mountant with DAPI (ThermoFisher).

### Confocal fluorescence imaging

Single point-scanning confocal microscopy was carried out on a Zeiss (Oberkochen, Germany) LSM 780 or LSM 710 microscope using a 40x/1.3 NA Fluar or 63x/1.4 NA Plan-apochromat oil immersion objective lens. The microscope was equipped with 405, 458, 488, 514, 561 and 633 nm laser lines. Multi point-scanning confocal microscopy was carried out on Visitech (Sunderland, UK) iSIM mounted on a Nikon (Tokyo, Japan) Ti2 microscope stand using a 100x/1.49 NA SR Apo TIRF oil immersion objective lens or a 60x/1.2 NA Plan Apo VC water immersion objective lens. The iSIM was equipped with 405, 445, 488, 561 and 640 nm laser lines and Hamamatsu (Hamamatsu, Japan) ORCA-Flash4.0 V3 sCMOS cameras. Filter ranges: green (500–545 nm), red (593–624 nm). Live-cell samples were heated to 37 ˚C and supplemented with 5% $CO_2$ using a microscope incubation chamber.

### Live-cell imaging and Fluorescence Recovery after Photobleaching (FRAP)

For live-cell staining of activated caspase-1, the supernatant was removed following inflammasome stimulation and replaced with DMEM supplemented with 10% (v/v) heat-inactivated FBS containing 0.5x reconstituted FAM-FLICA (ImmunoChemistry Technologies). For live-cell Golgi staining, LPS-primed iBMDMs were labeled with BODIPY TR ceramide (ThermoFisher, D7540) according to the manufacturer's protocol, prior to stimulation with nigericin. Cells were imaged as described above.

To quantify fluorescence of FAM-FLICA, BODIPY TR ceramide and ASC-mCerulean in the speck-forming area of a cell following stimulation (Fig. 3a), the cell outline was traced at each timepoint with the script Morph_ROI.ijm[60]. Fluorescence intensity within the speck-forming area and whole-cell fluorescence intensity and were measured at different time points. The graphs in Fig. 3a report what percentage of the whole-cell fluorescence mapped to the area of speck formation at two timepoints: at the beginning of the movie (1–3 min

after addition of nigericin) and at the time when speck formation reached completion (20–50 min after addition of nigericin).

For confocal FRAP, ASC-mCerulean iBMDMs were primed and induced as described above with the presence of caspase-1 inhibitor (Z-VAD-FMK). Live-cell FRAP imaging was performed on a Zeiss LSM 710 microscope equipped with a 63x/1.4 NA Plan-apochromat oil immersion objective lens and a 458 nm laser line for excitation and bleaching. The sample environment was heated to 37 ˚C and supplemented with 5% $CO_2$ using a microscope incubation chamber. ASC-mCerulean speck was photobleached with 100% laser power. Images were acquired at 3 s intervals. Images were collected at three pre-bleach timepoints and for 160 s post-bleaching. Movies were analysed in ImageJ/Fiji[47] using customized script FRAP_measure.ijm[60]. Briefly, measures were normalized to account for the general photobleaching caused by image acquisition as well as sample motion over time. Normalized fluorescence intensity measurements were obtained after background and bleaching corrections. The timeframe varied for experiments ($n = 8$). Each measurement was interpolated using the normalized fluorescence intensity measurements and imported to Python for plotting of the fluorescence intensity curve.

## Quantification of mitochondrial membrane potential

After 30 min of inflammasome stimulation as described above, iBMDMs were incubated with 20 nM TMRM (Life Technologies) for 15 min. iBMDMs were incubated for 3 h in 50 µM of the mitochondrial uncoupler carbonyl cyanide 3-chlorophenylhydrazone (CCCP; ThermoFisher, M20036). Control samples included: untreated; TMRM + ; TMRM + CCCP + ; TMRM + LPS+ iBMDMs. TMRM fluorescence was quantified by flow cytometry and data were acquired on Eclipse flow cytometer (Sony Biotechnology). Quantification was set at 100,000 for cell count and a spectrum window of FL3 (595BP) was used to detect TMRM signal. Data were analysed and visualized using FlowJo10.

## Cryo-ET sample preparation

Quantifoil Au 200-mesh finder grids (R2/1 or R2/2, Quantifoil Micro Tools) were glow discharged with a 30 mA current for 30 s with an Edwards S150B Sputter Coater. Grids were sterilized by UV irradiation for 10 min and immersed in PBS supplemented with 10 µg/ml fibronectin (Sigma) overnight in 8-well or coculture wells (ibidi). Grids were washed with PBS three times. Cells were seeded in 8-well or coculture wells (ibidi) and incubated overnight at 37 ˚C and 5% $CO_2$. Cells cultivated on grids were primed, induced with inflammasome stimuli and were plunge-frozen in liquid ethane using Leica EM GP2 cryo-plunger. Prior to plunging, 4 µl of cell culture medium was added to the cell side and backside blotting was applied for 6–8 s. The chamber conditions were maintained at 37 ˚C, 100% humidity during freezing. Grids were stored in liquid nitrogen.

## Cryo-fluorescent light microscopy (Cryo-FM) and cryo-focused ion-beam (FIB) milling

Grids were screened for cells with ASC/caspase-1 speck by light microscopy using a Leica EM Cryo CLEM microscope equipped with a cryo-stage, an ORCA-Flash4.0 V2 sCMOS camera (Hamamatsu Photonics) and a 50x/0.9 NA HCX PL APO cryo-objective lens. Montage acquisition of grids was performed with Leica LAS X software, while recording the following channels: green (L5 filter, 50 ms), far red (Y5 filter, 20 ms), and brightfield (50 ms). Z-stacks were recorded at 0.5 µm with step size 21 for each grid square to determine the best focus for the montage. The montage was completed by stitching the best focus image from the Z-stack with Leica LAS X.

Lamellae were prepared using a Scios Dual Beam FIB scanning electron microscope (SEM; ThermoFisher) equipped with a Quorum PP3010T cryo-stage. The milling protocol was adapted from a previously published method[33]. Grids were coated with organometallic platinum using the gas injection system (GIS; ThermoFisher) operated

at RT, for 8 s, at 12 mm working distance and 25˚ stage tilt. A first rough milling was performed at 25˚ stage tilt with a 30 kV ion beam voltage and 1 nA current until the lamella thickness reached 10 µm. The micro-expansion joints were applied to improve lamella stability[36] using the milling parameters listed above. The stage was tilted to 20˚ for subsequent milling steps. Rough milling steps were applied as following with 30 kV ion beam voltage: a 5 µm lamella thickness was reached with current 0.5 nA; 3 µm lamella thickness was reached with a 0.3 nA current; and 1 µm lamella thickness was obtained with a 0.1 nA current. Fine milling to a final lamella thickness of approximately 200 nm was performed with ion beam settings of 30 kV and 50 pA or 10 pA, or 10 kV and 23 pA or 11 pA.

Grids were loaded on a Linkam CMS196V cryo-stage, and lamellae were imaged on a Zeiss microscope equipped with Axiocam 503 mono and Colibri 7-illumination module R(G/Y) CBV-UV. Z-stacks of the lamellae were acquired to identify ASC/caspase-1 specks using a 200–300 nm step size. Only 3%–6% of the lamellae (one lamella out of 17–29 lamellae) retained fluorescent signal from ASC-mCerulean or FAM-FLICA after milling. A projected cryo-fluorescent image was used to correlate with an SEM image using the eC-CLEM plugin from ICY v1.9.5.1[37]. Registration between the lamella map and SEM image was achieved by transforming coordinates using edges or salient features of the lamella. The maximum projection Z-stack was then saved for correlation with SerialEM v3.8.0[38,39].

## Tilt-series acquisition and tomogram reconstruction

Grids with a fluorescent lamella were transferred to a 300 kV Titan Krios electron microscope (ThermoFisher) equipped with an energy filter (Gatan). Movies were acquired with a K2 or K3 direct electron detector with SerialEM[38,39]. Cryo-FM/EM correlation was then used to locate an area of interest for tilt-series acquisition. Briefly, a medium-magnification montage EM map (MMM) of the lamella was generated and a fluorescence map (FM) was loaded into SerialEM. The maps were correlated through recognition of geometric edges of the lamella. Transformation of coordinates yielded an overlay map which provided guidance for tilt-series acquisition. Tilt-series were collected at a nominal 42,000X, 33,000x or 26,000x magnification, resulting in pixel size 2.13 Å, 2.69 Å or 3.42 Å, over a tilt range of −60˚ to +60˚ with 1˚, or 2˚ increments, a total dose of 140–240 electrons A$^{-2}$ and a nominal defocus range of −4 to −8 µm. A dose-symmetric scheme was used[61]. A pre-processing script was used from SubTOM (by Dustin Morado [https://github.com/DustinMorado/subTOM]). Frames were aligned using IMOD[40]. Reconstruction was performed by weighted back-projection, and segmentation with IMOD. The MATLAB script deconv from Warp[62] was used for visualization. Filament tracing was performed manually with IMOD and UCSF Chimera[63]. Membrane segmentation was performed with TomoSegMemTV[64].

## Subtomogram averaging of ribosomes

302 ribosomes were manually picked in Dynamo[65]. Coordinates were converted to Warp format with the script dynamo2warp.py[66]. Subtomograms and corresponding contrast transfer function (CTF) models were reconstructed in Warp[62] with a box size of 44 pixels, a pixel size of 12 Å, and a particle diameter of 350 Å for normalization. Initial subtomogram alignment and averaging was performed in RELION v3.1[67] using a previously determined in situ cryo-ET structure of the mammalian 80S ribosome[68], low pass-filtered to 60 Å resolution, as the reference. This subtomogram average was used to perform template matching in Warp at 10 Å per pixel, with the addition of 808 particles. False positives were removed from template matching results. Subtomograms were exported from Warp to RELION for 3-D classification. A 3-D class with 1058 particles was selected without alignment and ref. 3.-D refinement was performed on the selected particles. The refined positions file was converted to RELION3.0 format with the script relion_star_downgrade.py[66]. Particles were exported at

with pixel size of 6 Å and refined using the average density as the reference. The resolution of the subtomogram average was calculated to be 23 Å during postprocessing in RELION v3.1 (using a Fourier shell correlation cut-off of 0.143). Ribosome subtomogram average volumes were mapped back into the 3-D segmented model with the script relionsubtomo2ChimeraX.py[69].

## Correlative fluorescence microscopy and electron tomography of resin-embedded cells

Correlative fluorescence microscopy and electron tomography (RT-CLEM) of resin-embedded cells was performed following a previously established method[45,70]. ASC-mCerulean cells were grown on carbon-coated 3 mm sapphire disks (Wohlwend GmbH) in two-well chambers (ibidi) for 24 h before priming and NLRPC-driven inflammasome stimulation as described above. Cells were stained with MitoView far-red and FLICA before high pressure freezing with an HPM100 high pressure freezing system (Leica Microsystems). Grids were imaged with a Leica EM Cryo CLEM cryo-fluorescence microscope to localize cells containing ASC-mCerulean specks. Freeze substitution was preformed using 0.008% uranyl acetate in acetone and embedded in Lowicryl HM20 (Polysciences) using an AFS2 (Leica Microsystems). Blocks were sectioned into 300 nm thin sections using a microtome (Leica Microsystems) equipped with a diamond knife (Diatome). The sections were collected on 200 mesh/300 mesh copper grids with carbon support (Agar Scientific). TetraSpeck 100-nm microspheres were diluted 1:100 in PBS and applied to the sections for use as fiducial markers for correlation. Sections were imaged on a Nikon Ti2 wide field microscope equipped with a Niji LED light source (Bluebox Optics), a x100/1.49 NA Apo TIRF oil immersion objective lens and a Neo sCMOS DC-152Q-C00-FI camera (Andor Technology). Filters: mCerulean (89006 filter set, Chroma Technology), fluorescein (49002 filter set; Chroma Technology), MitoView Far Red (49006 filter set; Chroma Technology). EM images were collected using a Tecnai F20 electron microscope (ThermoFisher) operated at 200 kV and a high-tilt tomography holder (Fischione Instruments, Model 2020). An image montage of regions of interest was acquired on a BM-Orius detector using TEM mode at 150–200 μm defocus using SerialEM v3.8.0[39] at a pixel size of 1.1 nm. The correlation between the montaged map and fluorescent map was performed by image transformation of registered fiducial markers in both image modalities following a previously described method[71]. Tilt-series were acquired from approximately −60 to +60 with 1° increment at a pixel size of 1.1 nm. Samples were rotated 90° to acquire dual axis tilt-series. Tomograms were reconstructed and visualized with IMOD[40].

## Subcellular fractionation and immunoblotting

Mitochondrial and cytosolic fractions were isolated using the Mitochondria/Cytosol Fractionation Kit (Abcam, ab65320). Briefly, 300 mg of ASC-mCerulean iBMDMs and WT iBMDMs were harvested after stimulation with LPS and nigericin. All centrifugation steps were performed at 4 °C. Cells were washed with ice-cold PBS and pelleted at 600 g for 5 min and resuspended in 1 ml 1x Cytosolic Extraction Buffer without protease inhibitors. After 10 min on ice, cells were homogenized on ice with 100 passes on pestle B. The cell lysate was centrifuged at 700 g for 10 min. The pellet ("PM pellet" in Fig. 5 and Supplementary Fig. 9), containing plasma membrane and any unlysed cells, was resuspended in RIPA buffer (50 mM Tris, 150 mM NaCl, 1% Triton X-100, 1 mM EDTA, 0.1% SDS). The supernatant was collected and centrifuged at 10,000 g for 30 min. The resulting supernatant was the cytosolic fraction ("Cytosol" in Fig. 5 and Supplementary Fig. 9). The pellet was resuspended in ice-cold 20 mM HEPES-KOH pH 7.5, 250 mM sucrose, 1 mM EDTA, and loaded onto a sucrose gradient prepared as described[72]. Briefly, the sucrose gradient was prepared by placing 1.5 ml of 60% sucrose buffer (60% sucrose, 20 mM HEPES-KOH pH 7.4, 1 mM EDTA) into an SW40 centrifuge tube, followed by 4.5 ml 32% sucrose buffer, 1.5 ml 23% sucrose buffer, and 1.5 ml 15% sucrose

buffer. The suspension was then centrifuged in an SW 40 Ti rotor at 99,004 g (28,000 rpm) for 1 h at 4 °C. A brown band containing the mitochondrial fraction ("Mitoch." In Fig. 5 and Supplementary Fig. 9) formed at the boundary between the 32% and 60% sucrose solutions and was extracted with a pipette. The cytosolic, mitochondrial and pellet fractions were immediately incubated at 95 °C for 10 min in 2x SDS-PAGE loading buffer and used for immunoblotting or stored at −20 °C.

The primary antibodies used for immunoblotting were as follows. Rabbit monoclonal anti-mouse GsdmD [EPR19828], 1:1000 dilution (Abcam, ab209845, RRID:AB_2783550); plasma membrane marker: rabbit monoclonal anti-mouse CD14, 1:1000 dilution (Abcam, ab221678, RRID:AB_2935854); cytosolic marker: mouse monoclonal anti-GAPDH, 1:5000 dilution (Proteintech, 60004-1-Ig, RRID: AB_2107436); mitochondrial marker: rabbit polyclonal anti-TOM20 FL-145, 1:500 dilution (Santa Cruz Biotechnology, sc-11415, RRID:AB_2207533).

## Quantifications and statistical analysis

Filament lengths were measured using IMOD[40] from five tomograms of ASC-mCerulean iBMDMs and one tomogram of WT iBMDMs labeled with FAM-FLICA. Histograms were plotted with MATLAB. Branching angles were measured with Fiji[47] and exported to MATLAB for plotting. Similarly, diameters of trans-Golgi-like vesicles were measured in Fiji and exported to MATLAB for plotting.

The ASC filament core diameter was determined from 8 Å-thick virtual tomographic slices from the ASC-mCerulean tomogram following neural network image restoration with cryoCARE[42]. Density line profiles 30 pixels in length (8 Å/pixel) were recorded perpendicular to the ASC filaments in Fiji. The density maxima corresponding to the core tube walls were identified in the density profiles ($n = 13$ for ASC-mCerulean) by calculating the first derivative of the smoothed density profile curve in GraphPad Prism v9.5.1 (the maxima were identified in the first derivative curve as x-axis intercepts with a negative slope). The density profiles were aligned by centering them on the midpoint of their respective tube-wall maxima positions. The aligned profiles were superimposed and used to fit a "sum of two Gaussians" function by non-linear regression in GraphPad Prism. The separation between the two maxima in the resulting function was taken as the filament core diameter.

Mitochondrial tomograms were loaded in Fiji[47] and ROI manager was used to measure morphological parameters, which were compared using Two-way ANOVA and plotted with GraphPad Prism. For the inner-to-outer membrane spacing, the distance between the nearest edges of the inner and outer membranes was measured. The cristae width was measured between the inner membranes of the cristae. The spacing between cristae were measured using the perpendicular distance between the internal border of the neighboring cristae. The inner-to-outer membrane spacing, cristae width and inter-cristae spacing were measured perpendicular to the membranes, evenly spaced for each tomogram, with thirty values per slice, for four slices per tomogram, from eight tomograms. The cristae apex angles were measured for every tip visible in ten slices per tomogram. Every value was plotted along with the average. If the end of a crista was flat, then the measurement of the two angles on either side was measured. Any measurement was only taken if the internal border of the membrane was clearly distinguishable from the matrix. All measurements were evenly spaced through each slice, to give a value representative of the whole slice, using all the mitochondria present in each image. All measurements were taken using 8 different tomograms: four control, two 30-min nigericin-treated, and two 60-min nigericin-treated mitochondria.

For measurement of the outer-mitochondrial membrane gap density, we calculated the density ratio as follows. A total of 21 virtual slices were selected in a region containing a visible outer-membrane gap. The average pixel gray value was calculated from circular areas

13.6 nm in diameter: three cytosolic areas, three inter-membrane space areas and an outer-membrane gap area. This was repeated for each of the 21 virtual slices. The ratio of average outer-membrane gap site to cytosolic gray value, and the ratio of average inter-membrane space site to cytosolic gray value were calculated for each virtual slice. The gray area ratio was then plotted with Igor (WaveMetrics, Inc.) to show the change of protein density in Z direction.

### Reporting summary
Further information on research design is available in the Nature Portfolio Reporting Summary linked to this article.

## Data availability
Representative cryo-ET data (electron tomograms) generated in this study have been deposited in the Electron Microscopy Data Bank (EMDB) under accession codes EMD-13585 and EMD-13586. The fluorescence recovery after photobleaching (FRAP) data, BODIPY TR ceramide fluorescence data, mitochondrial morphological parameter data, cryo_ET quantitative analysis data, and uncropped Western blots generated in this study are provided in the Source Data file. Source data are provided with this paper.

## Code availability
The Morph_ROI and FRAP_Measure scripts used in this study and instructions for use are available from the ASC_cryoET GitHub repository[60]. The relion_star_downgrade and dynamo2warp scripts used in this study and instructions for use are available from the dynamo2m GitHub repository[66]. The relionsubtomo2ChimeraX script used in this study and instructions for use are available from the subtomo2Chimera GitHub repository[69]. The subTOM suite used in this study and instructions for use are available from GitHub [https://github.com/DustinMorado/subTOM].

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

## Acknowledgements

We thank Dustin Morado for assisting in cryo-ET data collection and image processing. We thank Camilla Ventura Santos and Kunimichi Suzuki (MRC-LMB) for advice on subtomogram averaging with WARP. We thank Wanda Kukulski, Emma Jones, Robert Pickering, Maria Daly, Victoria L. Hale, Zunlong Ke, Panagiotis Tourlomousis, Long Chen, and Zhexin (Eric) Wang for advice and helpful discussions. We thank Eicke Latz for kindly providing the ASC-mCerulean iBMDM cells. We acknowledge the following core facilities at the MRC Laboratory of Molecular Biology for access, training, and support: Electron Microscopy, Flow Cytometry, Light Microscopy, and Scientific Computing. This work was supported by Senior Research Fellowships 101908/Z/13/Z and 217191/Z/19/Z from the Wellcome Trust to Y.M.; a PhD studentship from the China Scholarship Council and Cambridge Trust to Y.L.; and Investigator Award 108045/Z/15/Z from the Wellcome Trust to C.E.B.

## Author contributions

Conceptualization: Y.L., L.J.H., C.E.B., Y.M.; Formal Analysis: Y.L., H.Z., H.A., J.B., J.D.H., K.E.L.; Methodology: Y.L., C.E.B., Y.M.; Investigation:

Y.L., H.Z., J.D.H., A.C.B., C.H., L.J.H.; Visualization: Y.L., H.A., Y.M.; Funding acquisition: C.E.B., Y.M;. Project administration: C.E.B., Y.M.; Supervision: C.E.B., Y.M.; Writing – original draft: Y.L., C.E.B., Y.M.; Writing – review & editing: Y.L., C.E.B., Y.M.

## Competing interests

C.E.B. are Y.M. are consultants for Related Sciences LLC and have profits interests in Danger Bio LLC. C.E.B. is on the SAB of NodThera and Lightcast. The remaining authors declare no competing interests.
