## [Peer review file · Nature Communications]

REVIEWER COMMENTS

Reviewer #1 (Remarks to the Author):

In this revised manuscript, the authors have addressed my concerns. The new data on gasdermin D localization and activities is impressive. Before this manuscript can be considered complete, I have one final note. A recent study demonstrated that the previous conclusions of inflammasome protein localization to Golgi vesicles is incorrect.
Nat Immunol. 2023 Jan;24(1):30-41.

The authors of this study demonstrated that the marker used in prior work TGN38 to identify Golgi vesicles is actually present on endosomes. Bona fide markers of Golgi revealed no inflammasomes localization to this organelle and bona fide markers of endosomes validated that trans golgi markers accumulate on endosomes in the context of inflammasome activation.

The authors are encouraged to adjust the language used in the appropriate sections of this study to ensure the data presented in consistent with the most recent literature.

Reviewer #2 (Remarks to the Author):

Review of Manuscript NCOMMS-23-27811-T

Modis et al.: "Cryo-electron tomography of NLRP3-activated ASC complexes reveals organelle co-localization"

The authors have improved their manuscript concerning the previous version of the paper. In particular, the fractionation approach is interesting and indeed adds more to the biological novelty. However, the manuscript falls short of its potential to further improve the cryo-ET aspect.

Particularly, I still have objections regarding the following two broader aspects:

1) Technological

Reviewer 2 wrote: "This technological achievement is impressive." However, from a cryo-EM/ET perspective, it neither is nor was in 2021. Only very few tomograms were collected in total (based authors' response "Six tomograms were collected from ASC-mCerulean iBMDMs and one tomogram was collected from WT iBMDMs."; and from the paper: "all measurements were taken using 8 different tomograms: four control, two 30-min nigericin-treated, and two 60-min nigericin-treated"). Endorsing this paper would therefore invalidate years of work of groups (including mine), who have pushed the technology to be more than just "descriptive", and more than just "phenomenology".

I will illustrate, why this is a problem with a particularly striking example: In the previous version I asked how sure are the authors that the "density fluctuations" are indeed pores. All but one example fall in the direction of the missing wedge on the central y-axis. Again, I would ask how many of these "pores" were detected in total. How many per mitochondrion? Of course, lamellas capture only a minuscule part of the cell, but is it expected that there is only one pore per mitochondrion? Why would most of them be on the central y or x-axis?

Without suggesting foul play, is not very convincing if exactly these areas are missing in the examples

shown for “healthy/untreated” mitochondria (Fig, 4a).

To my original question, the authors responded with

“We have added additional examples of mitochondria with outer membrane gaps from multiple sites to Fig. 5a”

Maybe I have the wrong Fig. 5 or missed something else, but in my version the authors went from four to five examples, stretching the plural “additional exampleS” a bit thin ...

There is an additional argument to be had why performing manual mitochondria measurements in 2D is too biased (for state-of-the-art examples of unbiased, reliable measurements in 3D cryo-ET segmentations see <https://doi.org/10.1073/pnas.2209823119> and <https://doi.org/10.1083/jcb.202204093> for mitochondrial measurements)... However, in summary there is an overstatement of significance of the individual cryo-ET observations. This is not to say that the authors may not be correct. But in its present state, the cryo-ET part lacks the sustenance and rigor necessary to draw reliable conclusions.

2) Biological Standards

This is similar to my first argument, but not limited just cryo-ET. In Biology, conclusions should never be drawn from an $n = 1$ situation. Based on the authors’ response, only six tomograms have been acquired e.g. for the ASC-mCerulean iBMDMs. One for the wild type and five for their treatment conditions. It is not clear if any of these are biological or technological replicas. Today (and already 2021!), cryo-ET is mature enough to provide multiple replicas. It can and should be expected from tomographers, to provide repeats of biological experiments alongside the necessary statistical controls. I agree that taking more tomograms may not necessarily change the outcome of the findings. But it very well could. This is an important aspect to consider when weighing how much significance should be given to the cryo-ET data.

In summary, there are potential problems with the interpretation of the cryo-ET part of the paper from a technological and statistical standpoint. There is a low number of repeats if any, and statistics are derived from manual segmentations in 2D rather than 3D. Nonetheless, I want to highlight that I do recognize the work in general, and the additional work that has been added since the last version. Particularly, the fractionation experiments are interesting. Could a similar approach be used to add additional examples for the ASC-mCerulean iBMDMs and pores?

As to if the manuscript should be accepted, it is up to the editor to decide if the biological findings with a much-reduced impact of the cryo-ET observations are novel enough to support a publication in Nature Communications.

Reviewer #1 (Remarks to the Author):

In this revised manuscript, the authors have addressed my concerns. The new data on gasdermin D localization and activities is impressive. Before this manuscript can be considered complete, I have one final note. A recent study demonstrated that the previous conclusions of inflammasome protein localization to Golgi vesicles is incorrect. Nat Immunol. 2023 Jan;24(1):30-41.

The authors of this study demonstrated that the marker used in prior work TGN38 to identify Golgi vesicles is actually present on endosomes. Bona fide markers of Golgi revealed no inflammasomes localization to this organelle and bona fide markers of endosomes validated that trans golgi markers accumulate on endosomes in the context of inflammasome activation.

The authors are encouraged to adjust the language used in the appropriate sections of this study to ensure the data presented in consistent with the most recent literature.

We thank the reviewer for referring us to this new study by Zhang, de Matteis, Ricci and colleagues showing that the trans-Golgi marker TGN38 accumulates on endosomes during NLRP3 inflammasome activation due to disruption of ER-endosome contact sites, PI4P accumulation in endosomes and hence defective endosome to trans-Golgi trafficking.

We have modified all sections in the text relating to colocalization with Golgi markers to clarify that vesicles in which these markers accumulate following NLRP3 activation will include endosomal vesicles. The sections edited include the Introduction, in which the Zhang et al study is first cited (lines 56-62), the Results section (pp. 13-14, 16), and the figure legend heading for Fig. 3 legend (line 283). The subheading for the edited Results section has been changed to “Co-localization of ceramide-rich vesicles with ASC specks” (it was previously “Golgi vesicle localization in ASC specks”).

In addition was scanned the recent literature and added references to two journal articles reporting the formation of inactive cage-like decameric or dodecameric NLRP3 assemblies, based on single-particle averaging cryoEM reconstructions (Andreeva et al. 2021 and Hochheiser et al. 2022), along with a preprint reporting that NLRP3 can activate pyroptosis without colocalizing with TGN or MTOC markers and without forming cage-like oligomers (Mateo-Tortola et al. 2023).

Reviewer #2 (Remarks to the Author):

The authors have improved their manuscript concerning the previous version of the paper. In particular, the fractionation approach is interesting and indeed adds more to the biological novelty. However, the manuscript falls short of its potential to further improve the cryo-ET aspect.

We are pleased to note that Reviewer 2 found that the new subcellular fractionation data added in the previous revision increases the biological novelty of our study resulting in an improved manuscript. The reviewer's remaining concerns regarding the cryo-ET approaches are addressed below.

Particularly, I still have objections regarding the following two broader aspects:

1) Technological

Reviewer 2 wrote: "This technological achievement is impressive." However, from a cryo-EM/ET perspective, it neither is nor was in 2021. Only very few tomograms were collected in total (based authors' response "Six tomograms were collected from ASC-mCerulean iBMDMs and one tomogram was collected from WT iBMDMs."; and from the paper: "all measurements were taken using 8 different tomograms: four control, two 30-min nigericin-treated, and two 60-min nigericin-treated"). Endorsing this paper would therefore invalidate years of work of groups (including mine), who have pushed the technology to be more than just "descriptive", and more than just "phenomenology".

The Reviewer's efforts to develop and advance cryo-ET technology into a more quantitative and rigorous analytical tool are valid and laudable. In parallel to these efforts by valued members of the cryo-ET community, there is surely an important place for other researchers to apply existing cryo-ET technologies to obtain useful biological insights without necessarily pushing the technical boundaries of cryo-ET. Our goal in this study was to gain new insights on the ultrastructure of ASC puncta, which represent a key innate immune signaling hub. We make no claims in our manuscript of having advanced the state-of-the-art in cryo-ET. But our cryo-ET data provide the first glimpses of ASC puncta and mitochondria following NLRP3 activation. We show that ASC forms hollow filaments in the cell and provide quantitative measurements of ASC filament properties (outer and inner diameters, branching angles). We also quantify changes in four different parameters of mitochondrial morphology. We believe that on these merits our study will be of general interest to a broad readership.

Regarding the number of tomograms collected, for ASC speck imaging we collected 6 tomograms of ASC-mCerulean cells, of which 5 were biological replicates, and one tomogram of WT iBMDMs. Of these 9 tomograms, 5 were from lamellae in which ASC fluorescence was confirmed to be present by post-milling CLEM, and 4 tomograms were from purified mitochondrial fractions (newly collected for this revision). For mitochondrial morphology measurements, 8 additional tomograms were collected.

Our cryo-ET data collection throughput was limited by the high attrition rate at the FIB-milling stage. Indeed, in post-milling CLEM we found that 95% of our milled lamella the ASC punctum had been milled away during the milling procedure, or were otherwise unsuitable for data collection, e.g. due to ice accumulation or devitrification (see Fig. R3 below). In response to the reviewer's concern about the small number of tomograms collected, we collected four additional tomograms from purified mitochondrial fractions and obtained five additional examples of mitochondrial outer membrane gaps (see Fig. R2 below).

I will illustrate, why this is a problem with a particularly striking example: In the previous version I asked how sure are the authors that the “density fluctuations” are indeed pores. All but one example fall in the direction of the missing wedge on the central y-axis. Again, I would ask how many of these “pores” were detected in total. How many per mitochondrion? Of course, lamellas capture only a minuscule part of the cell, but is it expected that there is only one pore per mitochondrion? Why would most of them be on the central y or x-axis?

Without suggesting foul play, is not very convincing if exactly these areas are missing in the examples shown for “healthy/untreated” mitochondria (Fig, 4a).

Examination of the tomograms with the tilt axis overlaid showed that although the gaps shown in Fig. 5a were near the tilt axis, they did not coincide with it and other membrane features on the tilt axis were clearly visible without evidence of missing wedge artifacts. If missing wedge artifacts were present in our tomograms, these would have been visible along the entire tilt axis and not just at one mitochondrial membrane, but there is no evidence of this in our tomograms. The 5 tomogram slices shown in Fig. 5 are shown in Fig. R1 below with the position of the tilt axis overlaid. See also Fig. R2 below for 5 additional examples from four new tomograms collected for this revision, some of which are far from the tilt axis. Hence is overall no clear correlation between the positions of the tilt axis and outer membrane gaps. We have added a new supplementary figure to the revised manuscript, Fig. S8, showing all 10 tomograms with examples of mitochondrial outer membrane, with the position of the tilt axis overlaid for each tomogram.

We would like to note that the direction of the tilt axis is not the same in all tomogram slices shown in Fig. 5 – please refer to Fig. S8 for the tilt axis positions. We also show XY and XZ slices for each tomogram in Fig. S8 to demonstrate that none of our tomograms were flipped during reconstruction.

Outer membrane gaps were relatively rare – we only ever observed one gap per mitochondrion, but we note that in each case some of the mitochondrion in was milled away during FIB-milling.

Fig. R1 (Fig. S8a). Previously collected tomogram slices of FIB-milled cells showing mitochondrial outer membrane gaps with the position of the tilt axis overlaid (yellow line). Red lines indicate the x and y coordinates of the XZ and YZ sections, respectively.

To my original question, the authors responded with “We have added additional examples of mitochondria with outer membrane gaps from multiple sites to Fig. 5a”
 Maybe I have the wrong Fig. 5 or missed something else, but in my version the authors went

from four to five examples, stretching the plural “additional examples” a bit thin ...

In response to this comment we collected four new tomograms of mitochondria purified from cells stimulated with LPS and nigericin for 45 min. The tomograms of stimulated cells provide five additional examples of outer membrane gaps, none of which fall on the tilt axis. The new examples of outer membrane gaps are shown in the figure below and in a new supplementary figure in the revised manuscript (Fig. S8).

Fig. R2 (Fig. S8b). Additional examples outer membrane gaps in mitochondria from four newly generated cryo-ET tomograms showing (2nd to 5th panels from the left). The first panel from the left is an additional gap example from a previously collected tomogram. Panels 3-5 are from mitochondria purified from cells after stimulation with LPS and nigericin. The position of the tilt axis is shown in yellow. Red lines indicate the x and y coordinates of the XZ and YZ sections, respectively.

There is an additional argument to be had why performing manual mitochondria measurements in 2D is too biased (for state-of-the-art examples of unbiased, reliable measurements in 3D cryo-ET segmentations see <https://doi.org/10.1073/pnas.2209823119> and <https://doi.org/10.1083/jcb.202204093> for mitochondrial measurements)... However, in summary there is an overstatement of significance of the individual cryo-ET observations. This is not to say that the authors may not be correct. But in its present state, the cryo-ET part lacks the sustenance and rigor necessary to draw reliable conclusions.

As mentioned above, we have provided five additional examples of mitochondrial outer membrane gaps to increase the number of cryo-ET observations in response to this concern (see above).

We agree that measurements on tomograms in 2D (and also in 3D) can be prone to bias if the signal-to-noise ratio is low, but in the case of morphometric measurements on mitochondrial, the signal-to-noise ratio for the mitochondrial membranes is relatively high in our tomograms. Moreover, the expected morphology of mitochondria is well established by previous studies.

2) Biological Standards

This is similar to my first argument, but not limited just cryo-ET. In Biology, conclusions should never be drawn from an $n = 1$ situation. Based on the authors' response, only six tomograms have been acquired e.g. for the ASC-mCerulean iBMDMs. One for the wild type and five for their treatment conditions. It is not clear if any of these are biological or technological replicas. Today (and already 2021!), cryo-ET is mature enough to provide multiple replicas. It can and should be expected from tomographers, to provide repeats of biological experiments alongside the necessary statistical controls. I agree that taking more tomograms may not necessarily change the outcome of the findings. But it very well could. This is an important aspect to consider when weighing how much significance should be given to the cryo-ET data.

As mentioned above, we have collected four new tomograms, which support our original conclusions. In response to the reviewer's query about the number of biological replicates, we have created a workflow diagram that lists the numbers of FIB-milling sessions, lamellae, Krios sessions, and tomograms collected for the imaging of ASC puncta. For tomograms collected in the same Krios session, we specify the number of biological and technical replicates (Krios sessions in which a single tomogram was collected were biological replicates). The workflow diagram is shown below (Fig. R3) and was also included in the revised manuscript as a new supplementary figure, Fig. S3 in the revised version.

Fig. R3 (Fig. S3). Cryo-ET workflow and replicates supplementary figure showing the number of lamellae and tomograms generated for each cryo-CLEM biological replicate experiment performed.

In summary, there are potential problems with the interpretation of the cryo-ET part of the paper from a technological and statistical standpoint. There is a low number of repeats if any, and statistics are derived from manual segmentations in 2D rather than 3D. Nonetheless, I want to highlight that I do recognize the work in general, and the additional work that has been added since the last version. Particularly, the fractionation experiments are interesting. Could a similar approach be used to add additional examples for the ASC-mCerulean iBMDMs and pores?

We thank the reviewer for the suggestion to use mitochondrial subcellular fractions as a useful way to perform additional cryo-ET imaging. We used purified

mitochondrial fractions as suggested to obtain four new tomograms each containing a mitochondrial outer membrane gap (see Fig. R2 above and Fig. S8 in the revised manuscript).

As to if the manuscript should be accepted, it is up to the editor to decide if the biological findings with a much-reduced impact of the cryo-ET observations are novel enough to support a publication in Nature Communications.

As outlined above, we have added some new cryo-ET data in this revision. Our study provides the first views by cryo-ET of ASC puncta and mitochondria following NLRP3 activation. We show that ASC forms hollow filaments, provide quantitative measurements of ASC filament properties, and quantify changes in mitochondrial morphology. We believe that on these merits our study will be of general interest to a broad readership at *Nature Communications*.